# The Genetic Basis of Melanism in Abert’s Squirrel (*Sciurus aberti*)

**DOI:** 10.3390/ani14040648

**Published:** 2024-02-17

**Authors:** Lake H. Barrett, Dean Fraga, Richard M. Lehtinen

**Affiliations:** Biology Department, 931 College Mall, The College of Wooster, Wooster, OH 44691, USA; lbarrett23@wooster.edu (L.H.B.); dfraga@wooster.edu (D.F.)

**Keywords:** coloration, melanistic, melanocortin 1 receptor, agouti signaling protein, mutation, convergent evolution, single nucleotide polymorphism

## Abstract

**Simple Summary:**

Melanism, or the overexpression of melanin resulting in dark pigmentation, is fairly common among tree squirrels. The mutation that causes melanism has been discovered in two out of twelve species of tree squirrel that are known to have a melanistic phenotype. In this study, we identified the mutation that appears to cause melanism in a third species, Abert’s Squirrel (*Sciurus aberti*). The mutation is a previously unknown mutation apparently unique to Abert’s Squirrel, which suggests this mutation evolved independently from the other known mutations. Considering that three separate mutations are now known, it is likely there is a benefit for tree squirrels of having a dark, melanistic coat in certain environments.

**Abstract:**

Melanism is widespread in different taxa and has been hypothesized to provide adaptive benefits in certain environments. Melanism is typically caused by mutations in one of two regulatory genes: the Melanocortin 1 Receptor (*MC1R*) or the Agouti Signaling Protein (*ASIP*). Melanism has repeatedly evolved among tree squirrels and their relatives (tribe Sciurini) in at least 12 different species based on our review of the literature. The causal mutations for melanism have been characterized in two species so far. This study examines Abert’s Squirrel (*Sciurus aberti*), which has a melanistic morph whose genetic basis has not yet been established. We sequenced the *MC1R* and *ASIP* genes for five wild-type and seven melanistic *S. aberti* individuals to search for melanism-associated mutations. A novel single base pair mutation in the *ASIP* gene, unique to *S. aberti*, was found to be associated with melanism in the species, indicating that melanism in *S. aberti* evolved independently from other tree squirrels and thus represents an example of convergent evolution. The independent evolution of melanism in this species suggests that there is an adaptive advantage to the melanistic phenotype. The geographic range and habitat of *S. aberti* suggest possible benefits associated with thermoregulation, post-forest-fire camouflage, or other untested hypotheses.

## 1. Introduction

Coloration has many different important biological functions. A very short list of such functions includes mate choice [1], aposematism [2,3], mimicry [4], camouflage [5], thermoregulation [6], immune function [7], and metabolic rate [8]. As such, coloration can affect both an organism’s chance of survival and its reproductive success [1,9]. Advantages or disadvantages due to coloration have frequently been documented in organisms that possess different possible color phenotypes [10]. This phenomenon is known as a color polymorphism, which can be described as a species having multiple discrete coloration patterns [11]. These distinct coloration patterns are typically referred to as color morphs. Cases in which a particular color morph carries an adaptive advantage over another can result in selection for or against those morphs and lead to evolution. Because of this and the ease of observation due to the visible nature of the trait, color polymorphisms provide scientists the opportunity to monitor evolutionary processes by observing visible phenotypic variation over time and space [12].

Of the many striking color polymorphisms in the natural world, one of the most widespread is melanism [13]. The melanistic phenotype typically results from the overproduction of melanin, producing a phenotype with uniformly dark coloration. Melanistic individuals are seen in scores of different species, from insects to vertebrates [14,15]. The production of the pigment melanin in these organisms is likely to come at an additional cost to the organism [3] and would be expected to involve an adaptive benefit to the melanistic individual. While the adaptive significance of melanism is not always clear, melanistic phenotypes have been studied in many classic papers in genetics and evolution. For example, the peppered moth and industrial melanism provided an important early example of natural selection in action [16,17,18]. In another classic study of natural selection, melanistic phenotypes in the rock pocket mouse showed a strong correlation with dark-colored (volcanic) substrates compared to the light-colored phenotype found on adjacent lighter-colored rocks [19].

Of the more than 300 loci associated with melanin production [20], two genes with important and interacting roles have been repeatedly shown to be associated with melanistic phenotypes. These are the melanocortin-1 receptor (*MC1R*) gene and the agouti signaling protein (*ASIP*) gene. Mutations in the coding portions of either *MC1R* or *ASIP* have been tied to melanism in lizards [21], birds [22,23], and many different groups of mammals [24,25].

In squirrels, a twenty-four base pair deletion in *MC1R* as the causative element in melanism was first characterized by McRobie et al. [26] in the eastern grey squirrel (*Sciurus carolinensis*). Subsequently, the same twenty-four base pair deletion in *MC1R* was detected in western populations of the fox squirrel (*Sciurus niger*) [15]. Interestingly, melanistic individuals of the fox squirrel from the southeastern United States have a different mutation than the western populations, this time not in *MC1R* but a single base pair substitution in the *ASIP* gene [15,27]. These mutations both interrupt the eumelanin and phaeomelanin pathway, which determines coat color [27]. When functioning properly, the expression of the *MC1R* gene is promoted by the alpha-melanocyte-stimulating hormone (αMSH), which stimulates eumelanin production and is repressed by *ASIP*, resulting in the production of phaeomelanin [28]. When the 24 base pair deletion in the *MC1R* is present, a key glutamic acid is missing that is involved in αMSH binding [27]. This mimics αMSH binding and leads to the overproduction of eumelanin. When the single base pair substitution in *ASIP* is present, one of ten cysteines in *ASIP* that form an inhibitor cystine knot fold that binds to and inhibits *MC1R* is changed to a glycine, ultimately preventing the inhibition of *MC1R*, similarly allowing for overproduction of eumelanin [15,29]. 

The different mutations in *ASIP* and *MC1R* in different populations of fox squirrels clearly indicate multiple origins of the melanistic phenotype (i.e., convergent evolution [30]). However, the presence of the same mutation in both western populations of fox squirrels and the eastern grey squirrel is more difficult to interpret as there are multiple plausible causes. The same *MC1R* mutation could be the result of parallel evolution, ancestral polymorphism, or introgression between fox squirrels and grey squirrels. Parallel evolution is the independent evolution of the same phenotype via the same mutation [31]. Ancestral polymorphism is the retention of an ancient polymorphism that pre-dates the split between two or more taxa [32,33]. The latter explanation, introgression between different *Sciurus* species, proposes that ancient hybridization and backcrossing between the squirrel species introduced the allele from one species to the other. This explanation is favored by McRobie et al. [15], who hypothesized adaptive introgression from fox squirrels to grey squirrels. However, these three possibilities can be difficult to distinguish, and teasing out the evolutionary explanations for the origins of squirrel melanism is limited by a lack of information on the genetic basis of melanism in other closely related squirrel species. Based on a review of the literature, we are aware of no fewer than 12 squirrel species in the tribe Sciurini that exhibit melanism (Table 1), but so far, only *S. carolinensis* and *S. niger* have had the genetic basis of this trait characterized. In this study, we identify the genetic basis of melanism in an additional tree squirrel species: Abert’s squirrel (*Sciurus aberti*). 

## 2. Materials and Methods

*Sciurus aberti* (*Hesperosciurus aberti* of Abreau et al. [40]) has nine named subspecies and is geographically distributed in mountainous regions of the southwestern United States and northern Mexico [41]. However, only one of these subspecies, *S. a. ferreus*, has a known melanistic color morph (shown in Figure 1), and it is restricted to the northern edge of the species’ distribution [42]. 

Twenty-three tissue samples of *S. aberti* in total were obtained from the Denver Museum of Nature and Science (*n* = 12), the University of Colorado Boulder Museum (*n* = 3), the personal collection of Dr. John Koprowski (*n* = 6), and roadkill collected with the help of Colorado Parks and Wildlife (*n* = 2). Samples were obtained from the subspecies *Sciurus aberti ferreus* (*n* = 3 wild type and *n* = 12 melanistic), *Sciurus aberti aberti* (*n* = 7 wild type), and *Sciurus aberti kaibabensis* (*n* = 1 wild type).

Tissue samples were stored at −80 °C until DNA was extracted using a Qiagen DNEasy Blood and Tissue kit as described by the manufacturer (Qiagen Inc., Hilden, Germany). Purified DNA samples were quantified using a Thermo Scientific Nanodrop OneC (Thermo Fisher Inc., Waltham, MA, USA) and stored at −20 °C until use for PCR.

Primers for the *MC1R* gene of *S. aberti* were designed using the *MC1R* genes and adjacent genomic regions of *S. carolinensis* and *S. niger*, as the sequence of this region of the genome of *S. aberti* was not known. Wobble nucleotides were inserted for bases that differed between *S. carolinensis* and *S. niger* to account for potential differences in *S. aberti*. The forward primer used was (5′ TGG AYA CTG ACA GCT GGA GC 3′), and the reverse primer was (5′ TGG GAC SCG AAC KCT GGT CT 3′). This primer pair amplified an 1100 base pair fragment containing the entirety of the *MC1R* gene. A Qiagen Taq PCR Master Mix Kit (Qiagen Inc., Hilden, Germany) was used for all PCR reactions. PCR of the *MC1R* gene used the following thermocycler protocol: initial denaturation at 94 °C for two minutes, 34 cycles of 94 °C for 30 s, 62 °C for 30 s, and 72 °C for one minute, followed by a final extension at 72 °C for five minutes and held at 8 °C. The protocol was chosen after completing a temperature gradient to identify the ideal annealing temperature.

The known melanism mutation of *S. niger* is located in exon 4 of *ASIP*. Therefore, primers for the *ASIP* gene of *S. aberti* were targeted for this exon. The primers were designed using the *ASIP* gene and adjacent genomic regions of *S. carolinensis*, as this region of the genome of *S. aberti* was not known. For the *ASIP* gene, the forward primer (5′ CAG CAA GCA TGG ACA GCT CC 3′) and reverse primer (5′ AGG AAG CTK TGA GTG GAC GA 3′) were used. This primer pair amplified a 250 base pair fragment containing the entirety of exon 4 of the *ASIP* gene. PCR of the *ASIP* exon 4 used the following protocol: initial denaturation at 94 °C for two minutes, 34 cycles of 94 °C for 30 s, 59 °C for 30 s, and 72 °C for one minute, followed by a final extension at 72 °C for five minutes and held at 8 °C. The protocol was chosen after completing a temperature gradient to identify the ideal annealing temperature.

Following PCR, the product was verified using gel electrophoresis and purified using a Qiagen PCR Purification Kit. PCR products were sequenced in both directions at the Ohio State University Agricultural Research and Development Center (Wooster, Ohio). Bases were called, and contigs were built using MacVector version 17.0.5 (MacVector Inc., Apex, NC, USA). After clipping any low-quality reads, the DNA sequences were analyzed and aligned using a ClustalW alignment with default settings (open gap penalty of 15, an extended gap penalty of 6.7, delay divergent set to 30%, and transitions weighted). The multiple sequence alignments were then compared to search for known mutations or novel mutations consistently different between color morphs.

## 3. Results

### 3.1. Samples Sequenced

Out of the 23 total tissue samples available, a total of twelve samples were successfully amplified and sequenced for *MC1R* and a total of 13 were successfully amplified and sequenced for *ASIP*. The full *MC1R* genes were successfully sequenced for ten samples, five melanistic and five grey. A further two melanistic samples were partially sequenced, covering the region of the previously discovered 24 base pair deletion. The *ASIP* genes of six melanistic and seven grey morphs were successfully sequenced (Table 2).

### 3.2. MC1R Sequencing Results

The MC1R gene of *S. aberti* contained 15 base pair differences from that of *S. carolinensis* and 18 differences from *S. niger*. Notably, the 24 base pair deletion previously documented in *S. carolinensis* and *S. niger* was not present in melanistic morphs of *S. aberti*. Only one genotype of the *MC1R* gene was discovered in *S. aberti* (Figure 2). All melanistic morphs and all grey morphs had identical *MC1R* sequences.

### 3.3. ASIP Sequencing Results

The *ASIP* gene of *S. aberti* contained three different nucleotide polymorphisms from the corresponding gene of both *S. carolinensis* and *S. niger*. Two genotypes were discovered in the *ASIP* gene in *S. aberti* (Figure 3). Five grey morphs had the wild-type A-*ASIP*-1 genotype, characterized by thymine at base pair 370 in *ASIP* (145 in Figure 3) using the nomenclature of McRobie et al. [15]. This thymine is consistent with the wild-type *ASIP* exon 4 in both *S. niger* and *S. carolinensis*. In contrast, all six melanistic morphs had a cytosine at the same position, indicative of the A-*ASIP*-2 genotype. In addition to these results, two grey morphs were heterozygous for the wild-type allele and the mutant allele seen in melanistic morphs. This suggests that the mutant allele is recessive, and two copies are needed to produce a melanistic phenotype. The T370C substitution was only found in the subspecies *S. a. ferreus* and not in *S. a. aberti* or *S. a. kaibabensis*.

## 4. Discussion

Our work on *S. aberti* documents a single base pair mutation in the *ASIP* gene exclusively associated with melanistic individuals in this species. This mutation is a single nucleotide substitution at base pair 370 of *ASIP* (145 Figure 3), changing a thymine to a cytosine. This substitution results in a change in the amino acid sequence in the *ASIP* protein, turning a cysteine into an arginine. The amino acid change occurs in a critical area of the protein, where there is an inhibitor cysteine knot [29]. When properly assembled, this inhibitor knot prevents the function of the *MC1R* protein and, in doing so, regulates how much melanin is produced [15]. However, a change in one of the cysteines in this region disrupts the ability of the protein to fold up properly and function as an inhibitor [15]. This leads to the uncontrolled production of eumelanin and a melanistic phenotype. In the case of *S. aberti*, the swapping of a cysteine for an arginine causes this effect, resulting in the overproduction of eumelanin in the squirrels that have the mutation and produce a black phenotype. Since two grey morph individuals were heterozygous for the mutation, it would appear that the wild-type allele is dominant, and the mutant allele is recessive.

This novel mutation is different than the 24 base pair deletion in *MC1R* previously documented in *S. carolinensis* and western *S. niger* and the single base pair mutation in *ASIP* causing melanism in southeastern *S. niger*. However, the new mutation in *S. aberti* is very similar to a melanism-causing mutation that has been found in rats (*Rattus rattus*) [44]. This mutation occurs at the same base pair in the *ASIP* gene, but instead of a T→C (cysteine → arginine) mutation as found in *S. aberti*, it is a T→A (cysteine → serine) mutation [44]. Among other tree squirrels, the mutation found in *S. aberti* is similar to the mutation in the southeastern populations of *S. niger*. Melanistic individuals of *S. niger* in this region have a single base pair substitution located nine base pairs before the one reported here in *S. aberti* at base pair 136 in exon 4 [15]. The mutation in *S. niger* changes a glycine to a cysteine and disrupts the inhibitor region by the addition of an extra cysteine instead of the removal of a necessary one [15,29]. The effect remains the same, however, as both mutations and the mutation found in *R. rattus* appear to prevent the *ASIP* protein from inhibiting the *MC1R* protein. It is especially relevant that the same cysteine was substituted in a completely different species (*R. rattus*) [44] which also suggests that a change here is disruptive of protein function. Accepting this proposed mechanism of the mutation in *S. aberti*, it makes sense that the wild-type allele is dominant, and the mutant allele is recessive, as the heterozygous individuals that have a wild-type phenotype still possess one copy of the *ASIP* gene that can produce a functional inhibitor. While this novel mutation in *ASIP* is very likely to cause melanism in *S. aberti*, further experimental studies could confirm that this point mutation causes protein conformation and/or signaling pathway changes.

As noted above, the evolution of the same phenotype (in this case, melanism) in different evolutionary lineages has a number of possible explanations. These include convergent evolution (independently evolving the same phenotype by different mutations [30]), parallel evolution (independently evolving the same phenotype by the same mutation [31]), ancestral polymorphism (the alleles for the trait existed before a speciation event and were retained by descendent species [33]), or introgression (movement of alleles among species via hybridization and backcrossing; [45]). The unique nature and position of the *ASIP* mutation in *S. aberti* clearly indicates a unique origin for melanism in this species and strongly suggests convergent evolution as the mechanism. Parallel evolution, ancestral polymorphism, and introgression can all be ruled out since the mutation is not the same one that has been found in other species [15,27].

This is at least the third independent origin of this trait in the genus *Sciurus*. Alternatively, if we adopt the taxonomy of Abreu et al. [40], this results in one origin of melanism in *Hesperosciurus* (for *S. aberti*), one or two in *Parasciurus* (for *S. niger*) and zero or one in *Neosciurus* (for *S. carolinensis*), depending on whether we conclude that the 24 base pair deletion in *S. carolinensis* and western *S. niger* populations was introgressed from *S. niger* to *S. carolinensis* or vice versa [15]. However, a full understanding of the number of origins of this trait in tree squirrels more generally, and the mechanisms by which they were produced, will ultimately require characterization of the genetic basis of melanism in every species in the group that has the phenotype as well as a well-supported phylogeny of the entire group [40,46]. Our review of the literature suggests that at least nine other species have melanistic morphs for which the genetic basis is unknown (*S. aureogaster*, *S. flammifer*, *S. granatensis*, *S. oculatus*, *S. spadiceus*, *S. variegatoides*, *S. vulgaris*, *S. yucatanensis*, and *Tamiasciurus hudsonicus*; see Table 1). Of those, *S. vulgaris* has had its *MC1R* gene sequenced and is known to not have the 24 base pair deletion in the gene seen in *S. carolinensis* and *S. niger* [27]. Documenting the genetic basis of melanism in all these species will require no small amount of effort both in the field and in the lab; nonetheless, the fact that melanism has evolved repeatedly in this group suggests that it must have an adaptive benefit. 

In *S. aberti*, melanistic individuals are only known in the *ferreus* subspecies, which is found at the northernmost extent of its geographic range (in central and northern Colorado [42,47,48]). In these areas, the melanistic morph is common, representing the majority of individuals in some locations [42]. Since *S. aberti* is a mountain species that specializes in Ponderosa Pine [49], these northernmost populations experience long, cold winters [48]. However, the melanistic morph is not known from more southerly populations in New Mexico, Arizona, or Mexico [47]. Thus, the geographic pattern of the occurrence of the melanistic morph by itself might argue for a potential thermoregulatory adaptive benefit to melanism in *S. aberti*, which would represent an instance of selection varying among populations across the geographic range of a species (ex. [50]). Previous work on other tree squirrel species provides some support for this hypothesis [51,52]. However, studies in eastern fox squirrels (*S. niger*) have provided some evidence suggesting that the camouflage of melanistic individuals in post-wildfire environments may be an important selective advantage [53]. In dry western forests, forest fires were historically common and are now increasing in frequency [54]. Thus, a camouflage advantage seems possible as well. Another possibility that has been explored very little is the potential for pleiotropy, where the gene that causes melanism has multiple functions, as seen in wolves, where a black coat color has become common not because of an advantage to the coat color itself but as an advantage in the immune system tied to the black coat [7]. Still, additional hypotheses for the advantage of melanism include the possibility that the melanistic phenotype could convey a reproductive advantage and that sexual selection may be responsible for its frequency. All of these hypotheses for the adaptive benefit of melanism in *S. aberti* are testable, and future work should be directed in this area. 

## 5. Conclusions

The presence of a unique single base pair mutation in the Agouti Signaling Protein in melanistic Abert’s squirrels (*Sciurus aberti*) is very likely the causal genetic basis of melanism in this species. As this mutation is unknown among other species of related squirrels with a melanistic phenotype, this suggests independent (convergent) evolution of melanism in Abert’s squirrel. The recurring convergent evolution of this phenotype in tree squirrels and many other organisms suggests an adaptive benefit.

## Figures and Tables

**Figure 1 animals-14-00648-f001:**
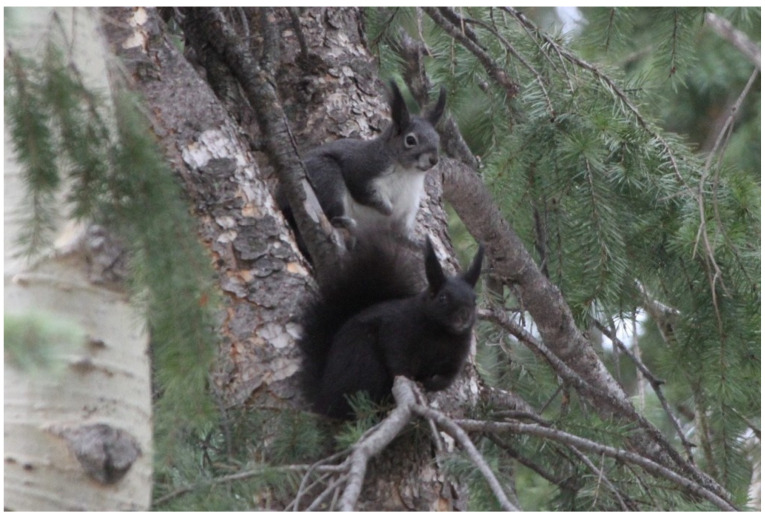
Wild-type and melanistic individual of *Sciurus aberti ferreus*. Above is a photo [43] depicting the wild-type (grey) morph and the melanistic morph of *S. aberti ferreus* from southwest Denver, Colorado.

**Figure 2 animals-14-00648-f002:**
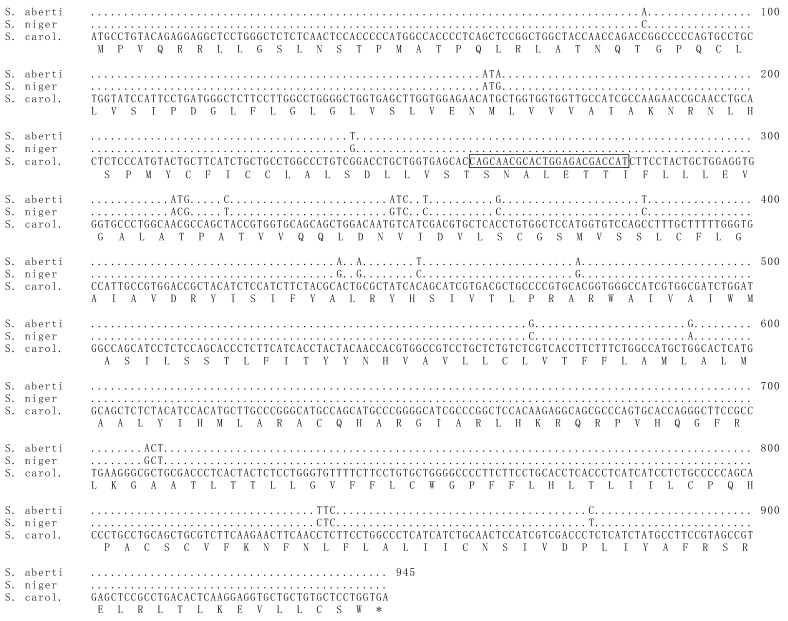
The *MC1R* gene of melanistic morphs of *S. aberti* did not show any notable mutations associated with melanism. Above is the *MC1R* gene from *S. aberti* aligned with those of *S. carolinensis* and *S. niger*. The full sequence of *S. carolinensis* is displayed, while positions in the sequence of *S. niger* and *S. aberti* containing exact matches are indicated by dots. Single nucleotide polymorphisms (SNPs) between the latter two species and *S. carolinensis* are shown using the one-letter code. The translated amino acid sequence from *S. carolinensis* is shown as one-letter abbreviation below the alignment. Those codons containing SNPs that result in amino acid changes between any of the three sequences are shown in their entirety. The position of the *MC1R* 24 base pair deletion seen in melanistic morphs of *S. niger* and *S. carolinensis* is boxed in the *S. carolinensis* sequence. Only one *S. aberti* sequence is shown, as all samples sequenced were identical.

**Figure 3 animals-14-00648-f003:**
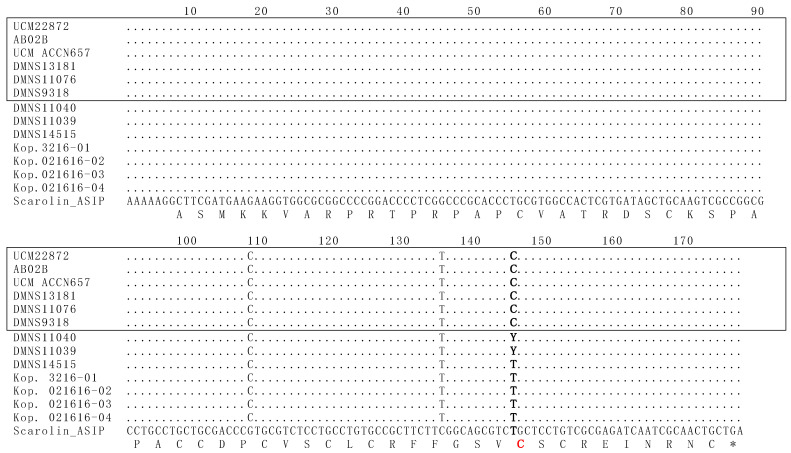
The *ASIP* gene of melanistic morphs of *S. aberti* shows a point mutation associated with melanism. The sequenced *ASIP* exon 4 fragment from melanistic morphs of *S. aberti* contains a polymorphism (T→C) at position 145, which corresponds to position 370 in the whole *ASIP* gene using the nomenclature of McRobie et al. 2019 [15]. The position in which the polymorphism occurs between the two color morphs is bolded. This substitution results in an amino acid change, converting a cysteine to an arginine at amino acid position C124 (C124R), which is indicated in red. The *S. carolinensis ASIP* exon 4 nucleotide sequence and its corresponding amino acid sequence are shown below for reference, with each amino acid letter indicated underneath the middle position of the corresponding codon. The black morphs are boxed, whereas the grey morphs are not boxed. Positions containing exact matches to the *S. carolinensis* sequence are indicated by dots, and single nucleotide polymorphisms (SNPs) are shown using the one-letter code. Sequence differences between the *S. carolinensis* and *S. aberti* species are indicated by the presence of the *S. aberti* nucleotide instead of a dot at positions 108 and 135. The Y at position 145 in this alignment indicates those two individuals were heterozygotes containing both alleles.

**Table 1 animals-14-00648-t001:** Tree squirrel (tribe: Sciurini) species that are documented as having a melanistic morph. This table shows all species in the tribe Sciurini, indicating those with a melanistic morph and the literature citation. The genetic mutation responsible for melanism is shown for species in which it has been uncovered. *MC1R* Δ24 represents a 24 base pair deletion in the *MC1R* gene, *ASIP* G361T represents a single base pair substitution in Exon 4 of *ASIP*, resulting in a Gly121Cys substitution in the *ASIP* protein, and *ASIP* T370C represents a single base pair substitution in Exon 4 of *ASIP* resulting in a Cys124Arg substitution in the *ASIP* protein.

Species	Melanism Exhibited	Source	Genetic Cause	Source
*Microsciurus alfari*	-	-	-	-
*Microsciurus flaviventer*	-	-	-	-
*Microsciurus mimulus*	-	-	-	-
*Microsciurus santanderensis*	-	-	-	-
*Rheithrosciurus macrotis*	-	-	-	-
*Sciurus aberti*	YES	[34]	*ASIP* T370C	This Study
*Sciurus aestuans*	-	-	-	-
*Sciurus alleni*	-	-	-	-
*Sciurus anomalus*	-	-	-	-
*Sciurus arizonensis*	-	-	-	-
*Sciurus aureogaster*	YES	[35]	Unknown	NA
*Sciurus carolinensis*	YES	[26]	*MC1R* Δ24	[26]
*Sciurus colliaei*	-	-	-	-
*Sciurus deppei*	-	-	-	-
*Sciurus flammifer*	YES	[36]	Unknown	NA
*Sciurus gilvigularis*	-	-	-	-
*Sciurus granatensis*	YES	[37]	Unknown	NA
*Sciurus griseus*	-	-	-	-
*Sciurus ignitus*	-	-	-	-
*Sciurus igniventris*	-	-	-	-
*Sciurus lis*	-	-	-	-
*Sciurus nayaritensis*	-	-	-	-
*Sciurus niger*	YES	[15]	*ASIP* G361T/*MC1R* Δ24	[15]
*Sciurus oculatus*	YES	[38]	Unknown	NA
*Sciurus pucheranii*	-	-	-	-
*Sciurus pyrrhinus*	-	-	-	-
*Sciurus richmondi*	-	-	-	-
*Sciurus sanborni*	-	-	-	-
*Sciurus spadiceus*	YES	[35]	Unknown	NA
*Sciurus stramineus*	-	-	-	-
*Sciurus variegatoides*	YES	[35]	Unknown	NA
*Sciurus vulgaris*	YES	[27]	Unknown	NA
*Sciurus yucatanensis*	YES	[35]	Unknown	NA
*Syntheosciurus brochus*	-	-	-	-
*Tamiasciurus douglasii*	-	-	-	-
*Tamiasciurus hudsonicus*	YES	[39]	Unknown	NA
*Tamiasciurus mearnsi*	-	-	-	-

**Table 2 animals-14-00648-t002:** Tissues acquired and sequenced. This table shows the samples collected and successfully genotyped as part of the study. Samples AB01B and 11076 were sequenced for the *MC1R* but did not yield a complete sequence of the gene. The GenBank accession number of each sample for the *MC1R* gene and *ASIP* gene is indicated, as is the museum accession number, if available. Samples are from collection localities in Colorado (CO) and Arizona (AZ). DMNS indicates the Denver Museum of Nature and Science, UCM indicates the University of Colorado Boulder Museum, Koprowski refers to the personal collection of Dr. John Koprowski and samples without a voucher were collected with the help of Colorado Parks and Wildlife.

Species	Collection Locality	ID #/Voucher Specimen	Color Morph	GenBank Accession # (*MC1R*)	GenBank Accession # (*ASIP*)
*Sciurus aberti aberti*	White Mountains, Navajo Co., AZ, USA	Koprowski 3216-01	Grey	NA	PP145380
*Sciurus aberti kaibabensis*	Kaibab Nat. Forest, Coconino Co., AZ, USA	Koprowski 3216-02	Grey	PP208811	NA
*Sciurus aberti aberti*	Mt. Graham, Graham Co., AZ, USA	Koprowski 021616-02	Grey	NA	PP145381
*Sciurus aberti aberti*	Mt. Graham, Graham Co., AZ, USA	Koprowski 021616-03	Grey	PP208812	PP145382
*Sciurus aberti aberti*	Mt. Graham, Graham Co., AZ, USA	Koprowski 021616-04	Grey	PP208813	PP145383
*Sciurus aberti ferreus*	County Road 126, Jefferson Co., CO, USA	DMNS 9318	Melanistic	PP208814	PP145371
*Sciurus aberti ferreus*	CR-83, Glenelk, Jefferson Co., CO, USA	DMNS 11039	Grey	PP208815	PP145377
*Sciurus aberti ferreus*	CR-83, Glenelk, Jefferson Co., CO, USA	DMNS 11040	Grey	PP208816	PP145378
*Sciurus aberti ferreus*	Keystone Drive, Jefferson Co., CO, USA	DMNS 11076	Melanistic	PP208817	PP145372
*Sciurus aberti ferreus*	CR-83, Glenelk, Jefferson Co., CO, USA	DMNS 13181	Melanistic	PP208818	PP145373
*Sciurus aberti ferreus*	Flagstaff Rd, Boulder Co., CO, USA	DMNS 14515	Grey	NA	PP145379
*Sciurus aberti ferreus*	Park Co., CO, USA	AB01B, no voucher	Melanistic	PP208819	NA
*Sciurus aberti ferreus*	Park Co., CO, USA	AB02B, no voucher	Melanistic	PP208820	PP145375
*Sciurus aberti ferreus*	Magnolia Road, Boulder Co., CO, USA	UCM ACCN657	Melanistic	PP208821	PP145376
*Sciurus aberti ferreus*	306 Little Ponderosa Way, Larimer Co., CO, USA	UCM 22872	Melanistic	PP208822	PP145374

## Data Availability

For DNA sequences generated in this study, see the GenBank accession numbers provided in Table 2.

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
