# Peer review of "The Genetic Basis of Melanism in Abert’s Squirrel (Sciurus aberti)"

_animals, 2024, doi:10.3390/ani14040648_

Round 1
Reviewer 1 Report
Comments and Suggestions for Authors
In this manuscript, the author attempts to uncover the genetic basis of melanism in Abert's squirrel (Sciurus aberti). This study contains some interesting findings and are valuable for the understanding of melanin production. However, the presented experimental evidence is insufficient and the experimental methods are too simplistic to effectively support the results, which fails to convince me.
1. The manuscript lacks direct experimental evidence regarding the phenotypes related to coat color. It would be beneficial for the author to visually present the melanism described in the selected S. aberti compared to other species, preferably through images.
2. The author performed PCR amplification and sequencing of the target sequences, but failed to describe the length of the amplified fragments and did not provide corresponding experimental images. Additionally, the use of a high-fidelity enzyme was not mentioned, which could ensure a lower error rate, particularly in sequencing, and enhance the credibility of the results.
3. The author's interpretation of the experimental results suggests the presence of point mutations in two genes, but the research methods employed are somewhat limited. There is a lack of experimental data supporting whether these point mutations cause changes at the gene or protein level, such as changes in the signaling pathway of the eumelanin and phaeomelanin synthesis or changes in protein conformation.
4. The labeling in Figure 2 is not intuitive and poses difficulties in understanding.
Based on the above concerns, I regret to inform the author that the manuscript is not suitable for publication in its current form. However, if the author can address these issues and provide additional experimental evidence and clarification, it may be reconsidered for publication.
Author Response
All reviewers comments are responded to in the attached document.

Reviewer 2 Report
Comments and Suggestions for Authors
See attached file.

Author Response

(The authors gave the same response as above.)

Reviewer 3 Report
Comments and Suggestions for Authors
This article is of undoubted interest to specialists in the fields of genetics, ecology and evolutionary ecology and is devoted to the molecular genetic comparison of individuals of melanistic phenotypes of three polymorphic fur-colored species of the genus Sciurus based on two regulatory genes. The authors have strictly proved the different genetic nature of the occurrence of melanistic morphs in the S. alberti compared with S. carolinensis and S. niger. There is every reason to agree with the authors' conclusion that melanistic coloration in S. alberti was evolutionarily formed as a parallel convergent effect and may be due to adaptive color changes in some individuals in the populations of the species. The obtained result complements the existing few examples of adaptive polymorphism and is important for biology and zoology in general. I have no comments on the applied molecular methods and the general analysis of the material. The presentation of the results also does not deserve comments, the text is presented logically and in extremely clear language. However, I have a small formal comment on the design of the work. In my opinion, it is unacceptable not to give the names of the tables, referring the reader only to the explanations in the notes to them. I believe that the same remark applies to the absence of the name Figure 2. After making these minor corrections, this article deserves to be published in Animals.
Author Response

(The authors gave the same response as above.)

Reviewer 4 Report
Comments and Suggestions for Authors
This work is devoted to the search for the genetic basis of melanism in the Abert's squirrel. Authors sequenced two genes for which the effect on the recurrence of dark pigmentation was previously shown. The experiment is clearly constructed, the conclusions are confirmed by the results. I have only a small number of editorial comments.
1) It is necessary to write the abbreviations of genes in italics;
2) In some places, the Latin names of species also remained not in italics;
3) At the first mention of each Latin name, it is necessary to add the author and the year of the description (see also lines 225, 262-264);
4) Tables and figures have no names;
5) I would like to arrange the drawings in a single style. It is convenient if, instead of nucleotides identical to the reference, there are dots (and not as in Figure 2 – at the bottom under each column). For the second picture, by analogy with the first one, you can add the names of amino acids. It seems to me that these small changes will simplify the visual perception.
Author Response

(The authors gave the same response as above.)

Round 2
Reviewer 1 Report
Comments and Suggestions for Authors
I have carefully reviewed the revised version and would like to provide the authors with further feedback. Firstly, I appreciate the authors have made in addressing the concerns raised during the initial review. The revised manuscript has improved in terms of clarity and organization. However, I would like to suggest a few additional references that could enhance the overall strength and credibility of your study.
1. Matrosova VA, et al. (2019) Phylogenetic relationship and variation of alarm call traits of populations of red‐cheeked ground squirrels (Spermophilus erythrogenys sensu lato) suggest taxonomic delineation. Integr. Zool. 14(4):341-353.
2. Jin L, Liao WB, & Merilä J (2022) Genomic evidence for adaptive differentiation among Microhyla fissipes populations: Implications for conservation. Divers. Distrib. 28(12):2665-2680.
3. Long J, Zheng Y, Luan X, & Liao WB (2022) Genomic differentiation with isolation by distance along a latitudinal gradient in the spotted-leg treefrog Polypedates megacephalus. Integr. Zool. 11.
4. Lei ZX, et al. (2023) Genomic evidence for climatic adaptation in Fejervarya multistriata. Divers. Distrib.:14.
5. ZAMORA‐CAMACHO FJ (2022) Sex and habitat differences in size and coloration of an amphibian's poison glands match differential predator pressures. Integr. Zool. 17(5):764-776.
Please consider incorporating these references into your revised manuscript to further support your arguments and strengthen the scientific foundation of your study.
Overall, I commend you on the improvements made to the manuscript and appreciate your attention to the reviewers' comments. I believe that incorporating the suggested references will enhance the quality and impact of your research.
